# New Insights on Molecular Autopsy in Sudden Death: A Systematic Review

**DOI:** 10.3390/diagnostics14111151

**Published:** 2024-05-30

**Authors:** Luca Tomassini, Giulia Ricchezze, Piergiorgio Fedeli, Massimo Lancia, Cristiana Gambelunghe, Francesco De Micco, Mariano Cingolani, Roberto Scendoni

**Affiliations:** 1International School of Advanced Studies, University of Camerino, 62032 Camerino, Italy; luca.tomassini@unicam.it; 2Department of Law, Institute of Legal Medicine, University of Macerata, 62100 Macerata, Italy; mariano.cingolani@unimc.it (M.C.); r.scendoni@unimc.it (R.S.); 3School of Law, Legal Medicine, University of Camerino, 62032 Camerino, Italy; piergiorgio.fedeli@unicam.it; 4Forensic Medicine, Forensic Science and Sports Medicine Section, Department of Medicine and Surgery, University of Perugia, 06132 Perugia, Italy; massimo.lancia@unipg.it (M.L.); cristiana.gambelunghe@unipg.it (C.G.); 5Research Unit of Bioethics and Humanities, Department of Medicine and Surgery, Università Campus Bio-Medico di Roma, 00128 Roma, Italy; f.demicco@unicampus.it

**Keywords:** molecular autopsy, sudden unexpected death, sudden death in young people (SUDY), genetic testing, forensic examination, sudden death

## Abstract

Sudden unexpected deaths often remain unresolved despite forensic examination, posing challenges for pathologists. Molecular autopsy, through genetic testing, can reveal hidden causes undetectable by standard methods. This review assesses the role of molecular autopsy in clarifying SUD cases, examining its methodology, utility, and effectiveness in autopsy practice. This systematic review followed PRISMA guidelines and was registered with PROSPERO (registration number: CRD42024499832). Searches on PubMed, Scopus, and Web of Science identified English studies (2018–2023) on molecular autopsy in sudden death cases. Data from selected studies were recorded and filtered based on inclusion/exclusion criteria. Descriptive statistics analyzed the study scope, tissue usage, publication countries, and journals. A total of 1759 publications from the past 5 years were found, with 30 duplicates excluded. After detailed consideration, 1645 publications were also excluded, leaving 84 full-text articles for selection. Out of these, 37 full-text articles were chosen for analysis. Different study types were analyzed. Mutations were identified in 17 studies, totaling 47 mutations. Molecular investigations are essential when standard exams fall short in determining sudden death causes. Expertise in molecular biology is crucial due to diverse genetic conditions. Discrepancies in post-mortem protocols affect the validity of results, making standardization necessary. Multidisciplinary approaches and the analysis of different tissue types are vital.

## 1. Introduction

Sudden unexpected deaths (SUD) are devastating events for families of the deceased victim and occur worldwide [1]. In the majority of countries, such incidents call for forensic examination to determine the cause of death, but this frequently remains unresolved [2].

Such deaths occur without notable warning signs. Sudden death among seemingly healthy people (from infants to adults) poses a dilemma for forensic pathologists, law enforcement personnel, and society at large [3,4].

Sudden death is a frequent event, particularly in the young population, accounting for 10% of all deaths in the age group of 1 to 22 years, with dramatic effects on families [5].

Sudden death may result from a combination of conditions including certain types of arrhythmias such as long QT syndrome; these pathologies are not apparent at autopsy, since they do not leave macroscopic evidence [6].

When cause of death is unidentified, genetic testing of DNA extracted from post-mortem blood and tissues (molecular autopsy) may help to identify a likely cause of death. Regardless of genetic testing considerations, all families in which a sudden unexplained death has occurred require targeted and standardized clinical testing to identify relatives who may be at risk of having the same inherited heart disease and therefore predisposed to an increased risk of sudden cardiac death. Optimal care for families affected by sudden cardiac death thus requires dedicated and appropriately trained staff within a specialized multidisciplinary cardiac genetic clinic [6,7].

In light of this, the gene testing approach shows promise and could be widely integrated into forensic practice. Pressing priorities include gathering and broadening existing datasets, alongside establishing new genetic databases through molecular autopsy. These efforts would streamline the genetic methodology for routine application [2,8].

The objective of this investigation was to conduct a literature review aimed at evaluating the proportion of SUD cases clarified through molecular autopsies. The available literature on molecular autopsy in various types of sudden death was reviewed, delineating the investigative methodology, the types of studies in which genetic analysis has been conducted, and their actual utility and efficacy in autopsy practice.

In comparison to similar studies, where the frequency with which molecular autopsy resolves cases of sudden death or the genes most commonly associated with sudden death have been investigated, the present work aimed to focus primarily on the methodological aspects of individual studies, potentially highlighting elements of incompleteness that may render these studies challenging to apply in practical and theoretical contexts [2,9,10].

## 2. Materials and Methods

This systematic review followed the Preferred Reporting Items for Systematic Reviews and Meta-analyses (PRISMA) reporting guidelines. The study protocol was registered with PROSPERO under registration number CRD42024499832. The PRISMA checklist is available as a Appendix A Appendix A.

A systematic literature search was conducted on PubMed, Scopus, and Web of Science to identify studies published in English between 2018 and 2023.

The objective of this study was to examine the existing literature concerning molecular autopsy in diverse instances of sudden death. This involved detailing the investigative approach, identifying the study types in which genetic analysis has been performed, and assessing their practical application and effectiveness in autopsy procedures, all summarized from the available literature.

The generic free-text search terms were: (“Cardiac” [All Fields]) AND (“molecular” [All Fields] OR “moleculars” [All Fields]) AND (“autopsied” [All Fields] OR “autopsy” [MeSH Terms] OR “autopsy” [All Fields] OR “autopsies” [All Fields] AND (“sudden” [All Fields] OR “unexpected” [All Fields] AND “death” [All Fields]). Filters applied were: “Case Reports”, “Classical Article”, “in the last 5 years”, “Humans”, and “English”.

Two researchers independently searched PubMed, Scopus, and Web of Science for studies, while three other researchers checked whether the selected articles met the inclusion criteria. The following data were recorded from the chosen studies: authors, country in which the study was performed, date of publication, genes investigated, heart diseases, and types of tissue samples analyzed. The resulting documents underwent further filtering by inspection of the language, title, abstract, methods, and keywords. Those finally selected for analysis had to respect the following inclusion criteria: Original articles or case studies;Non-violent deaths;Post-mortem genetic testing;Study of human tissues;Studies in which the purpose was to clarify or formulate a postmortem forensic diagnosis;Cases inclusive of sudden death in infants below 1 year of age (SIDS) and sudden death in infants between 1 and 5 years of age (SUDI) and SCD and SUDY, without specification of the number of each category, were included.

Non-inclusion and exclusion criteria were:
Studies focused exclusively on SIDS—sudden death in infants below 1 year of age;Studies focused exclusively on SUDI—sudden death in infants between 1 and 5 years of age;Genetic studies performed on tissues taken from living people;Genetic studies on animal tissues;Genetic studies whose results were obtained in vitro, even if a database of post-mortem material from earlier studies was used to obtain them;Genetic studies aimed at identifying the DNA or RNA of infectious agents;Scientific works aimed at improving the technical approach to the use of the genetic testing method without the task of clarifying or making a forensic medical diagnosis;Review articles, systematic reviews, meta-analyses, practical recommendations, monographs, and commentary articles on previous research.

The study was designed according to the PRISMA recommendations, as shown in Figure 1. Descriptive statistics were applied to the data. The selected articles were analyzed for the scope of molecular autopsy study and the tissues used. In addition, information on the countries of publication and journal names and dates were collected and analyzed.

The data collection process included both the study selection and data extraction. 

As mentioned previously, three researchers independently assessed whether the articles had titles or abstracts that met the inclusion criteria, and any disagreements were resolved by achieving consensus. Two researchers performed the data extraction, which was then reviewed by two other researchers and subsequently reconfirmed by an additional pair of investigators. 

## 3. Results

In total, 1759 publications from the preceding 5 years met the search criteria. 

A total of 30 duplicate articles were excluded. After careful consideration of whether studies met the primary inclusion criteria, a further 1645 publications were excluded, leaving 84 full-text articles. Publications aimed at clarifying or making a forensic medical diagnosis were selected, and another 47 articles were excluded from the sample. The remaining 37 full-text articles therefore fully met the inclusion criteria for the review. The article selection process is summarized in Figure 1.

Various types of samples were used for the genetic investigations. Blood was the most commonly employed sample type: it was used in 15 studies. Additionally, six mixed samples (frozen tissues, frozen/fresh blood, post-mortem paraffine embedded tissue) were used. However, the sample type was not specified in nine cases. Other types of samples included frozen heart tissue (1), shock-frozen renal tissue (1), heart paraffin sample (1), lymphocytes/leukocytes (2), heart tissue (1) and formalin-fixed and paraffin-embedded left ventricular tissue (1). The distribution of withdrawal types is summarized in Figure 2.

Regarding the country of the included studies, six (16%) were performed in Italy, four (11%) in Germany, four (11%) in Switzerland, three (8%) in Spain, two (5%) in Canada, two (5%) in Denmark, two (5%) in Japan, two (5%) in the United Kingdom, two (5%) in the USA, and the remaining in Austria (3%), Belgium (3%), China (3%), Colombia (3%), France (3%), Italy-Spain (3%), New Zealand (3%), Norway (3%), Portugal (3%), and Tunisia (3%).

In total, we analyzed: one case–control study; nine retrospective studies; thirteen case reports; one epidemiological study; nine research articles; one observational, transversal, and retrospective study; one observational study; one retrospective observational study; and one brief research report.

Excluding the thirteen case reports, of the remaining twenty-four studies, only six (25%) provided a detailed section within the methods describing the statistical analysis including the statistical tests used. Among the studies that described the statistical analysis, two used the χ^2^ test, four used Fisher’s exact test, one employed logistic regression, one utilized pathway enrichment analysis, one used the interquartile range, one employed the linear effect model, one used the likelihood ratio test, one utilized the Mann–Whitney U test, and one employed the *t*-test. Among these studies, four used more than one statistical test in combination, while two studies used a single test.

Out of the total, twelve studies exclusively focused on cases of SCD; five cases examined SCD, and sudden unexpected death in young population (SUDY); three cases considered SCD, SUDI, and SUDY; one case involved instances of SIDS, SUDY, and SUDI; and thirteen studies investigated cases of SUDY.

In twenty cases, cardiac pathology was not specified, while in nine cases, the studies focused on arrhythmic pathologies. Six studies centered around structural cardiac pathologies, and two studies discussed other pathologies (coronary pathology and myocarditis). The type of study, the gene(s) examined, the pathology under investigation (if specified), the characteristic of the sample utilized, the number of individuals examined, the results, and observations concerning individual articles are summarized in Table 1. Appendix A reports all data extrapolated from the studies considered.

Regarding the mutations examined, 37 studies identified a total of 51 mutations. Specifically, mutations in the PKP2 and PPA2 genes were observed in two cases each. ACTN2, CACNA1C, CALR3, DSG2, KCNE1, KCNJ2, TCAP, and TTN were all involved in a single case. The MYH7 gene presented seven distinct mutations. KCNH2 and KCNQ1 were involved in five cases. LMNA, SCN5A, TNNI3, and TNNT2 each had three different mutations. The MYBPC3 gene present five identified mutations. MYH7 was involved in seven cases, and RYR2 was involved in five cases. Mutations classified as ‘pathogenic’ were observed in 24 cases. In 12 cases, mutations were classified as ‘pathogenic/likely pathogenic’. Finally, mutations classified as ‘likely pathogenic’ were identified in 15 cases.

The individual mutations and their pathogenicity in relation to the studies are summarized in Table 2. Table 2 is summarized graphically in Figure 3.

## 4. Discussion

Standard autopsy does not always detect an individual’s cause of death, and this is a common occurrence in cases of sudden death. 

Recently, there has been a significant increase in attention toward diagnostic protocols for sudden death related to the cardiovascular system (SCD). These protocols provide a structured framework for molecular genetic analysis and the interpretation of results, enabling the identification of pathogenic or likely pathogenic genetic variants that may be responsible for SCD. This multidisciplinary approach involves forensic specialists, cardiologists, and molecular geneticists, enhancing the understanding of the causes of sudden death and facilitating the implementation of appropriate preventive and therapeutic measures for at-risk family members [3,48]. 

The diagnostic pathway in cases of sudden death has also been the focus of recommendations that provide a flowchart for the diagnostic process to be followed [49,50].

Furthermore, the adoption of standardized protocols for autopsy and post-mortem genetic analysis contributes to the establishment of national guidelines and the promotion of better clinical practice in SCD management [51,52].

From the perspective of the diagnostic pathway, the cause of death, at least in a portion of unresolved cases, may be revealed using molecular biology and genetic testing methods [53,54]. The pathologies underlying these conditions are often dominantly inherited, and precise diagnosis through molecular autopsy can reveal the genetic risk for first-degree relatives, enabling timely preventive interventions. Therefore, the preservation of tissue samples for DNA analysis is crucial [55]. A thorough review of the selected scientific literature allowed us to outline the evolution of molecular autopsy practices over the previous five years. For decades, forensic medicine has grappled with the challenge of identifying the origins of non-violent deaths in cases lacking discernible morphological alterations in the body [56].

The analysis conducted encompassed a variety of studies including case–control studies, retrospective cohorts, and a series of reports and observational studies. Furthermore, the review considered a heterogeneous array of conditions such as sudden cardiac death (SCD) and sudden death in young people (SUDY). 

The pathologies considered also exhibited heterogeneity, as there was significant variability concerning the treated cardiac pathologies; in some cases, the exact pathology was not specified, while in others, the focus was on arrhythmias and structural anomalies. The diversity of the sample highlights the complexity of the conditions studied in the context of sudden cardiac related death. It is worth noting that in 20 studies, the pathology under investigation was not specified, making it impossible to determine which sudden death anomaly the study referred to.

Excluding studies where pathologies and/or mutations were not identified, in our examination of genetic mutations associated with various cardiovascular pathologies, we uncovered several significant findings across different genes. These mutations shed light on the underlying genetic factors contributing to conditions such as HCM, DCM, ARVD, LQTS, BrS, and CPVT.

For instance, mutations in genes like ACTN2, CACNA1C, and CALR3 have been identified as pathogenic, implicating them in conditions like HCM, DCM, LQTS, familial HCM, and ARVC. Similarly, mutations in DSG2 and KCNE1 have shown likely pathogenicity, associated with ARVD and LQTS, respectively.

Significant pathogenic mutations have been found in genes like KCNH2, KCNJ2, LMNA, MYBPC3, MYH7, PKP2, SCN5A, and RYR2, each linked to specific cardiovascular conditions. These include LQTS, HCM, ARVD, BrS, and CPVT, underscoring the diverse genetic landscape underlying these disorders.

Furthermore, mutations in genes like TTN, TNNI3, and TNNT2 have been associated with a spectrum of cardiomyopathies, reflecting the complexity of genetic influences on cardiac function and structure.

Regarding the frequency of different etiologies based on the examined genes, we assessed whether there were differences in the distribution of pathologies among different genes. 

For instance, mutations associated with LQTS were more frequent in the KCNH2 and KCNQ1 genes, in accordance with the literature data [57], while mutations associated with DCM were found to be more common in the FLNC, LMNA, MYBPC3, and TTN genes. Additionally, mutations associated with CPVT appeared to be more frequent in the RYR2 gene. However, it should be noted that most studies did not express a clear judgment regarding a significant portion of the identified mutations; we therefore summarized them in Table 2 as ‘likely pathogenic’. In several cases, specific genes were identified as ‘likely pathogenic’ due to specific variants, considering that ‘likely pathogenic’ indicates pathogenicity up to 90% [58]. As seen, in several cases, the significance of mutations was not fully understood. Consequently, the diagnosis of death could only be provided presumptively, albeit with a high statistical probability of pathogenicity. This underscores the importance of considering mutations in a relative context across studies.

Finally, these results highlight the diversity of sample types used in genetic investigations, with a preference for blood as the primary sample. However, it is important to note that in many cases, the type of sample used was not specified, which could influence the accuracy and validity of the obtained results. Furthermore, the presence of a variety of sample types such as lung, renal, cardiac tissue, and lymphocytes/leukocytes suggests that a comprehensive and multidisciplinary approach is necessary in analyzing the genetic causes of sudden death. This diversity of samples may allow for a more thorough and comprehensive assessment of the possible underlying genetic causes in cases of sudden death, thereby contributing to an improved understanding and management of such events.

In other words, in the field of molecular autopsy, there still exists a wide variety of variables that pose a significant obstacle to the definitive understanding of certain sudden deaths. This issue represents a diagnostic challenge for the involved professionals, which can only be overcome through the development of new tools by the scientific community.

In this regard, molecular autopsy may be limited by the lack of standardized international diagnostic protocols for molecular autopsy, which vary considerably from one region to another (between countries and even within the same country), evolving from those already proposed [3,58]. Furthermore, this review highlights the diversity in the types of samples used in genetic investigations, which shows a lack of uniformity in the protocols for collecting and analyzing post-mortem samples. This could impact the accuracy and validity of the obtained results and could complicate comparisons between studies conducted in different contexts.

It is believed that the mutations identified in various studies may significantly influence the accuracy of DNA forensic identification. This is particularly important for the stability and predictability of the genetic markers used. Such mutations could lead to erroneous matches or mismatches, compromising the reliability of the investigations. Furthermore, considering these data, updating forensic genetic databases becomes essential to account for these new mutations and maintain data relevance [4,7].

It should be noted that the prevalence of pathological mutations may be much higher in individuals with sudden cardiac arrest, a factor that needs to be taken into account [57].

The presence of new mutations may require the recalibration of statistical analyses and genetic match probabilities. It is important in this regard to consider the impact of these mutations on the confidence levels in the results and interpretation of genetic evidence.

The implications also extend to forensic methodologies, which may require adjustments to incorporate these new findings. For example, the adoption of advanced sequencing technologies can help better detect and understand genetic mutations. Additionally, the continuous updating of standard operating procedures and adequate training for forensic experts is necessary.

Finally, from a legal and ethical standpoint, it is crucial to consider the implications of genetic mutations on laws and individual rights. This includes privacy protection and informed consent for the use of genetic data for forensic purposes.

From a statistical perspective, among the 24 original articles, which were heterogeneous in type, it is evident that the vast majority did not include a “statistical analysis” section in the methods where the statistical tests used were reported. In fact, only six studies clearly discussed this aspect inin their respective Methods sections. Most of these studies had a qualitative nature, leading the authors to believe they did not have sufficient material to perform statistical comparisons of the results.

### Study Limitations

We decided to include in our discussion only variations reported as ‘pathogenic’ or ‘likely pathogenic’. Therefore, we intentionally left out those reported as variants of unknown significance (VUS). We also decided not to include new variants that were not supported by sufficient scientific evidence.

In many studies, information was not provided on pathologies associated with mutations. Accordingly, we searched the NCBI ClinVar database for these.

It was not possible to identify which genes were investigated in all studies. In one case, the data were inaccessible because of the authors’ opposition, while in many others, the panel type of screened genes was mentioned, but the names of these genes were not given.

Finally, in many studies, a distinction was not made between sudden cardiac deaths (SDC) and sudden deaths in young people (SUDY). Therefore, based on the existing literature, we adopted the following definitions: SUDY—sudden death in individuals between 5 and 35 years of age and SCD—sudden death in adults older than 35. Whenever it was not possible to discriminate between these, we reported them simply as SCD.

It is noted that certain studies included cases of SIDS and SUDI alongside SCD and SUDY without specifying their numbers or providing a statistical assessment of the distribution of mutations for each category; however, these studies have nonetheless been included, as specified in Table 1.

## 5. Conclusions

The complexity of sudden death causes is highlighted by the need for molecular investigations, which are crucial in determining causes when they are not clearly identifiable through standard morphological examinations. However, the broad spectrum and complexity of the genetic conditions involved necessitate a deep level of knowledge and expertise in molecular biology and genetics for accurate diagnosis. A lack of uniformity in post-mortem diagnostic protocols is evident both across different countries and within individual national contexts. This disparity may influence the validity and reliability of the obtained results, emphasizing the importance of standardizing molecular autopsy protocols and post-mortem genetic investigations.

The diversity in the types of samples used in post-mortem genetic investigations underscores the importance of a multidisciplinary and comprehensive approach in assessing the genetic causes of sudden death. Analyzing a variety of tissues may allow for a more detailed and comprehensive evaluation of possible underlying genetic causes, thereby contributing to an improved understanding and management of such situations. Limitations in interpreting genetic mutations, especially when the significance of mutations is not fully understood, pose a challenge in the accurate interpretation of the results of post-mortem molecular analysis. 

## Figures and Tables

**Figure 1 diagnostics-14-01151-f001:**
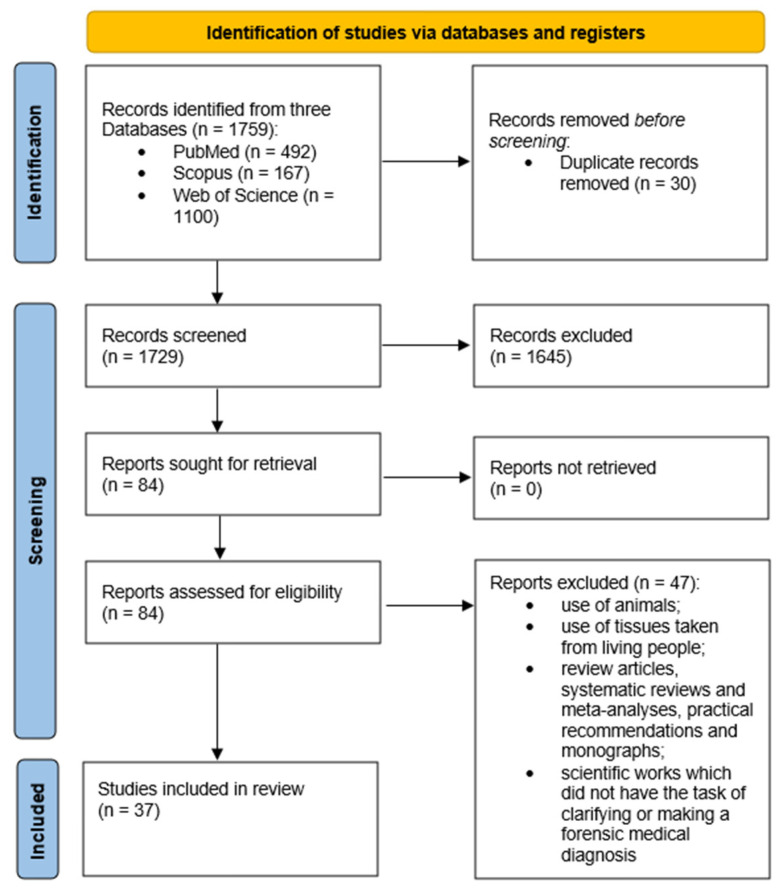
Descriptive diagram of the paper selection process.

**Figure 2 diagnostics-14-01151-f002:**
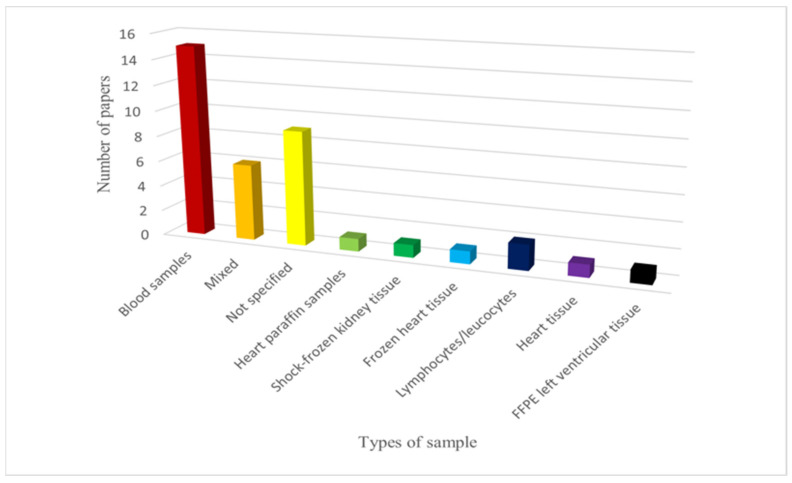
Types of samples used in the various articles. FFPE: formalin-fixed and paraffin-embedded left ventricular tissue.

**Figure 3 diagnostics-14-01151-f003:**
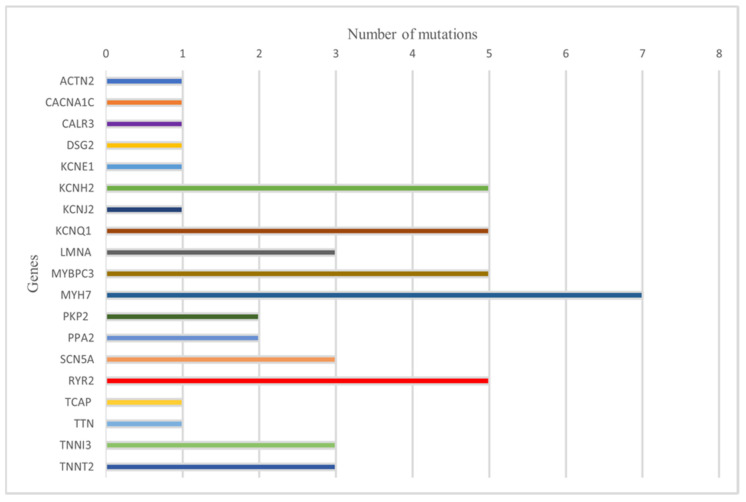
Summary graph on the frequency of mutations.

**Table 1 diagnostics-14-01151-t001:** Synthesis table of the studies analyzed. SCD: sudden cardiac death; SUDY: sudden unexpected death in young population; SUDI: sudden death in infants between 1 and 5 years of age; SIDS: sudden death in infants below 1 year of age.

Authors and Year of Publication	Gene(s) Examined	Heart Disease	Sample Number and Type of Death
Zhen X. et al. (2023) [11]	1 gene (CAG)_n_ repeat polymorphism within Androgen Receptor (AR) gene	Coronary heart disease	564 healthy controls and 182 cases of SCD
Neubauer J. et al. (2022) [12]	393 cardiovascular and metabolic disease genes	Not specified	39 cases of SCD
Alhassani S. et al. (2018) [13]	30 genes (AKAP9, ANK2, CACNA1C, CACNB2, CASQ2, CAV3, DSC2, DSG2, DSP, GPD1L, HCN4, JUP, KCNE1, KCNE2, KCNE3, KCNH2, KCNJ2, KCNJ5, KCNJ8, KCNQ1, NKX2.5, PKP2, RANGRF, RYR2, SCN1B, SCN3B, SCN4B, SCN5A, SNTA1, TMEM43)	Not specified	A case of SUDY and his family
Ariza J.A. et al. (2022) [14]	4834 clinically relevant genes	Cardiac channelopathy	A case of SUDY
Marey I. et al. (2020) [15]	15 genes (MYH7, MYBPC3, TNNT2, TNNI3, MYL2, PKP2, DSP, DSG2, LMNA, TTR, and five major sarcomeric genes in DCM)	Not specified	35 cases of SCD
Siskind T. et al. (2022) [16]	94 genes (ABCC9, ACTC1, ACTN2, AKAP9, AKAP10, ANK2, ANKRD1, ARHGAP24, BAG3, BCAT1, CACNA1C, CACNA2D1, CACNB2, CALM1, CAML2, CASQ2, CAV1, CAV3, CDKN1A, CRYAB, CSRP3, CTF1, DES, DPP6, DSC2, DSG2, DSP, DTNA, EMD, FHL2, FLRT2, GATAD1, GLA, GPD1L, HAND1, HCN4, JUP, KCNA5, KCND3, KCNE1, KCNE1L, KCNE2, KCNE3, KCNE4, KCNH2, KCNJ2, KCNJ5, KCNJ8, KCNQ1, LAMA4, LAMP2, LDB3, LMNA, MYB, MYBPC3, MYH6, MYH7, MYL2, MYL3, MYLK2, MYOZ2, MYPN, NEBL, NEXN, NOS1AP, PC3, PKP2, PLN, PRKAG2, RANGRF, RBM20, RyR2, SCN10A, SCN1B, SCN2B, SCN3B, SCN4B, SCN5A, SGCD, SNTA1, STRN, TAZ, TCAP, TGFB3, TMEM43, TMPO, TNNC1, TNNI3, TNNT2, TPM, TPM1, TRPM4, TTR, and VCL)	Not specified	5 cases of SCD and 6 cases of SUDY
Clemens D.J. (2020) [17]	1 gene (TRDN)	Triadine knockout syndrome (TKOS)	258 cases of SUDY
Marzialiano N. et al. (2019) [18]	60 genes (ACTC1, ACVRL1, APOB, BAG3, BMPR2, BRAF, CACNA1C, CASQ2, DES, DMD, DSC2, DSG2, DSP, ELN, EMD, ENG, FBN1, FLNC, GATA4, GLA, JAG1, JUP, KCNE1, KCNE2, KCNH2, KCNJ2, KCNJ8, KCNQ1, KRAS, LAMP2, LDLR, LDLRAP1, LMNA, MYBPC3, MYH7, MYL2, MYL3, NF1, NKX2-5, PKP2, PLN, PRKAG2, PCSK9, PTPN11, RAF1, RBM20, RYR2, SCN1B, SCN5A, SOS1, SOS2, TAZ, TGFBR2, TMEM43, TNNC1, TNNI3, TNNT2, TPM1, TTN, and TTR)	Hypertrophic cardiomyopathy and heterozygous familial hypercholesterolemia	A case of SUDY
Beccacece L. et al. (2023) [19]	The DNA was genotyped for about 720,000 genetic markers	Not specified	30 cases of SCD
Iglesias M. et al. (2021) [20]	From 194 to 380 genes	Not specified	31 cases of SUDI, SUDY and SCD (unspecified number)
Larsen M.K. et al. (2020) [21]	104 genes (ABCC9f, ACTC1a,g, ACTN2l, AKAP9, ANK2, ANKRD1l, BAG3, CACNA1Cc, CACNA1D, CACNB2, CALM1h,n, CALM2h,n, CALM3h,n, CALR3, CASQ2, CAV3c, CRYAB, CSRP3a, CTF1, DESk, DMD, DPP6, DSC2, DSG2a, DSPa, DTNA, EYA4, FHL2, FKTN, GAAa, GJA5i,j, GLAl, GPD1Ld, HCN1e, HCN4e,f,g, ILK, JPH2f, JUP, KCNA5, KCND3f, KCNE1f, KCNE2f, KCNE3f, KCNE4f, KCNE5f, KCNH2f,o, KCNJ2f,o, KCNJ5f, KCNJ8f,h, KCNQ1e,h,o, LAMA4, LAMP2, LDB3g,l, lHCM, LMNAf,g,k, MOG1, MYBPC3a,g, MYH6a,e, MYH7a,g, MYL2, MYL3, MYLK2, MYOZ2, MYPNl, NEBL, NEXNl, NPPA, PKP2b, PLNk,l, PRDM16g, PRKAG2f,m, PSEN1, PSEN2, RANGRFb, RBM20, RPS7, RPSA, RYR2f,k, SCN1Bf,i, SCN2Bb, SCN3Bf, SCN4Bf, SCN5Aa,c,e,f,g,h,i,j,k, SDHA, SGCD, SLC22A5, SLC25A4, SNTA1, STARD3, TAZg, TCAPl, TGFB3, TMEM43, TMPO, TNNC1l, TNNI3l, TNNT2a,g, TPM1a,g, TRPM4i,j, TTNa,k, and VCLa)	Hypertrophic cardiomyopathy (HCM), dilated cardiomyopathy (DCM), arrhythmogenic right ventricular	70 cases of SUDI, SUDY and SCD
Girolami F. et al. (2022) [22]	174 genes (ABCC9, ABCG5, ABCG8, ACTA1, ACTA2, ACTC1, ACTN2, AKAP9, ALMS1, ANK2, ANKRD1, APOA4, APOA5, APOB, APOC2, APOE, BAG3, BRAF, CACNA1C, CACNA2D1, CACNB2, CALM1, CALR3, CASQ2, CAV3, CBL, CBS, CETP, COL3A1, COL5A1, COL5A2, COX15, CREB3L3, CRELD1, CRYAB, CSRP3, CTF1, DES, DMD, DNAJC19, DOLK, DPP6, DSC2, DSG2, DSP, DTNA, EFEMP2, ELN, EMD, EYA4, FBN1, FBN2, FHL1, FHL2, FKRP, FKTN, FXN, GAA, GATAD1, GCKR, GJA5, GLA, GPD1L, GPIHBP1, HADHA, HCN4, HFE, HRAS, HSPB8, ILK, JAG1, JPH2, JUP, KCNA5, KCND3, KCNE1, KCNE2, KCNE3, KCNH2, KCNJ2, KCNJ5, KCNJ8, KCNQ1, KLF10, KRAS, LAMA2, LAMA4, LAMP2, LDB3, LDLR, LDLRAP1, LMF1, LMNA, LPL, LTBP2, MAP2K1, MAP2K2, MIB1, MURC, MYBPC3, MYH11, MYH6, MYH7, MYL2, MYL3, MYLK, MYLK2, MYO6, MYOZ2, MYPN, NEXN, NKX25, NODAL, NPPA, NRAS, PCSK9, PDLIM3, PKP2, PLN, PRDM16, PRKAG2, PRKAR1A, PTPN11, RAF1, RANGRF, RBM20, RYR1, RYR2, SALL4, SCN1B, SCN2B, SCN3B, SCN4B, SCN5A, SCO2, SDHA, SEPN1, SGCB, SGCD, SHOC2, SLC25A4, SLC2A10, SMAD3, SMAD4, SNTA1, SOS1, SREBF2, TAZ, TBX20, TBX3, TBX5, TCAP, TGFB2, TGFB3, TGFBR1, TGFBR2, TMEM43, TMPO, TNNC1, TNNI3, TNNT2, TPM1, TRDN, TRIM63, TRPM4, TTN, TTR, TXNRD2, VCL, ZBTB17, ZHX3, ZIC3)	Not specified	14 cases of SCD
Neubauer J. et al. (2021) [23]	244 (ABCC8, ABCC9, ACAD9, ACADM, ACADS, ACADVL, ACTA2, ACTC1, ACTN2, ACVRL1, ADAMTS10, AGL, AKAP9, ALG10, ALMS1, ANK2, ANKRD1, ASCL1, ATP5F1E, BAG3, BDNF, BMPR1B, BMPR2, BRAF, CACNA1C, CACNA2D1, CACNB2, CALM1, CALM2, CALM3, CALR3, CAMK2G, CASQ2, CAV1, CAV3, CAVIN4, CBL, CDH2, CHRM2, CLCA2, COA5, COL3A1, COL5A1, COL5A2, COL6A1, COL6A2, CPT1A, CPT2, CRYAB, CSRP3, CTF1, CTGF, CTNNA3, DCHS1, DES, DLG1, DMD, DMPK, DNAJC19, DNM1L, DOLK, DPP6, DSC2, DSG2, DSP, DTNA, ECE1, EDN3, EFEMP2, ELN, EMD, ENG, ETFA, ETFB, ETFDH, EYA4, FBN1, FBN2, FGF12, FHL1, FHL2, FHOD3, FKRP, FKTN, FLNA, FLNC, FXN, G6PC, GAA, GATA4, GATA5, GATA6, GATAD1, GDNF, GJA1, GJA5, GJD4, GK, GLA, GLB1, GLRA1, GPD1L, GUSB, HADH, HADHA, HADHB, HCN2, HCN4, HEY2, HFE, HMGCL, HMGCS2, HRAS, HTR2C, ILK, JPH2, JUP, KCNA5, KCND2, KCND3, KCNE1, KCNE2, KCNE3, KCNE5, KCNH2, KCNJ2, KCNJ5, KCNJ8, KCNK17, KCNQ1, KLF10, KRAS, LAMA4, LAMP2, LDB3, LMNA, LRP5, LRRC10, LZTR1, MAOA, MAP2K1, MAP2K2, MED12, MED23, MOG, MRPL3, MT-TI, MT-TL1, MYBPC3, MYH11, MYH6, MYH7, MYL2, MYL3, MYLK, MYLK2, MYO6, MYOM1, MYOZ2, MYPN, NEBL, NEXN, NKX2-5, NOS1AP, NOTCH1, NPPA, NPPA, NRAS, PDLIM3, PDSS2, PHOX2B, PKP2, PLEKHM2, PLN, PPA2, PPP1R13L, PRDM16, PRKAG2, PRKG1, PSEN1, PSEN2, PTPN11, RAB3GAP1, RAF1, RANGRF, RBM20, RET, RYR2, SCN10A, SCN1B, SCN2B, SCN3B, SCN4B, SCN5A, SCO2, SDHA, SEMA3A, SGCD, SHOC2, SKI, SLC22A5, SLC25A10, SLC25A3, SLC37A4, SLC4A3, SLC6A4, SLMAP, SMAD3, SMAD9, SNTA1, SOS1, SYNE1, SYNE2, TAZ, TBX1, TBX20, TBX3, TBX5, TCAP, TGFB2, TGFB3, TGFBR1, TGFBR2, TMEM43, TMEM70, TMPO, TNNC1, TNNI3, TNNI3K, TNNT2, TP63, TPM1, TRDN, TRIM63, TRPM4, TRPM7, TSFM, TSPYL1, TTN, TTR, TXNRD2, VCL, XK, and ZNF365)	Not specified	45 cases of SCD
Scheiper-Welling S. et al. (2022) [24]	93 genes with known cardiac associations	Arrhythmic heart disease	56 cases of SUDY
Fadoni J. et al. (2022) [25]	40 genes (MYBPC3, MYH7, TNNI3, TNNT2, ACTC1, TPM1, MYL2, MYL3, MYH6, TNNC1, VCL, CAV3, MYLK2, JPH2, CSRP3, ANKRD1, DES, ACTN2, MYL4, NEXN, CRYAB, DSG2, HSPB1, HSPD1, MYO6, GPD1L, KCNE2, NME1, MYC, POMC, SCN5A, TP53, ACAD9, GAA, PRKAG2, LAMP2, NDUFS1, RAF1, SCO2, and SCL25A4)	Hypertrophic cardiomyopathy (HCM)	16 cases of SUDY and SCD
Martínez-Barrios E. et al. (2023) [26]	113 genes (ABCC9, ACTA2, ACTC1, ACTN2, AKAP9, ANK2, BAG3, CACNA1C, CACNA1G, CACNA1H, CACNA1I, CACNB2, CASQ2, CAV3, CHRM2, COL3A1, CRYAB, CSRP3, CTF1, DES, DMD, DMPK, DSC2, DSG2, DSP, ECE1, EMD, EN1, EYA4, FBN1, FHL2, FKTN, GJA7, GLA, GPD1L, HCN1, HCN2, HCN4, ILK, JPH2, JUP, KCNA4, KCNA5, KCND2, KCND3, KCNE1, KCNE2, KCNE3, KCNH2, KCNJ2, KCNJ3, KCNJ5, KCNK4, KCNQ1, LAMA4, LAMP2, LDB3, LMNA, MYBPC3, MYH6, MYH7, MYL2, MYL3, MYLK2, MYOZ2, MYPN, NEBL, NEXN, NOS1AP, NOTCH1, NPPA, NUP155, PDLIM3, PHOX2A, PHOX2B, PKP2, PLN, PRKAG2, PSEN1, PSEN2, RBM20, RET, RYR2, SCN10A, SCN1B, SCN2B, SCN3B, SCN4B, SCN5A, SGCA, SGCB, SGCD, SIRT3, SLC25A4, SLC6A4, SLC8A1, SLMAP, SNTA1, TAZ, TCAP, TGFB3, TGFBR1, TGFBR2, TLX3, TMEM43, TMPO, TNNC1, TNNI3, TNNT2, TPM1, TTN, and VCL)	Not specified	51 cases of SIDS, SUDI and SUDY (unspecified number)
Tuveng Jon M. et al. (2018) [27]	5 genes (KCNQ1, KCNH2, SCN5A, KCNE1, KCNE2	Not specified	A case of SUDY
Kraoua L. et al. (2012) [28]	Whole genome	Hypertrophic/dilatated cardiomyopathy	A case of SUDY
Gélinas R. et al. (2019) [29]	184 genes (AARS2, ABCC6, ABCC9, ACAD9, ACADVL, ACTA1, ACTA2, ACTC1, ACTN2, AGK, AGL, AKAP9, ALMS1, ALPK3, ANK2, ANO5, APOA1, BAG3, BRAF, CACNA1C, CACNB2, CALM1, CALM2, CALM3, CALR3, CAPN3, CASQ2, CAV3, CBL, CDH2, COX15, CPT2, CRYAB, CSRP3, CTNNA3, DBH, DES, DMD, DNAJC19, DOLK, DSC2, DSG2, DSP, DTNA, DYSF, EEF1A2, ELAC2, EMD, ENPP1, EPG5, ETFA, ETFB, ETFDH, FBXO32, FHL1, FKRP, FKTN, FLNC, FOXD4, FOXRED1, FXN, GAA, GATA5, GATA6, GATAD1, GBE1, GFM1, GLA, GLB1, GMPPB, GTPBP3, GUSB, HADHA, HAND1, HCN4, HFE, HRAS, ISPD, JPH2, JUP, KCNA5, KCNE1, KCNE2, KCNH2, KCNJ2, KCNJ5, KCNQ1, KRAS, LAMA2, LAMP2, LARGE, LDB3, LMNA, LRRC10, LZTR1, MAP2K1, MAP2K2, MLYCD, MTO1, MYBPC3, MYBPHL, MYH6, MYH7, MYL2, MYL3, MYL4, MYOT, MYPN, NDUFAF2, NEXN, NF1, NKX2-5, NOS1AP, NRAS, NUP155, PCCA, PCCB, PKP2, PLEC, PLEKHM2, PLN, PNPLA2, POMT1, PPA2, PPP1CB, PRDM16, PRKAG2, PTPN11, RAF1, RASA2, RBCK1, RBM20, RIT1, RMND1, RRAS, RYR2, SALL4, SCN10A, SCN1B, SCN3B, SCN5A, SCNN1B, SCNN1G, SCO2, SDHA, SELENON, SGCA, SGCB, SGCD, SGCG, SHOC2, SLC22A5, SLC25A20, SLC25A4, SMCHD1, SOS1, SOS2, SPEG, SPRED1, TAB2, TAZ, TBX20, TBX5, TCAP, TECRL, TGFB3, TMEM43, TMEM70, TNNC1, TNNI3, TNNI3K, TNNT2, TOR1AIP1, TPM1, TRDN, TRIM32, TRPM4, TSFM, TTN, TTR, VCL, VCP, VPS13A, and XK)	Not specified	A case of SUDY
Takahashi Y. et al. (2023) [30]	72 genes (ABCC9, ACTC1, ACTN2, AKAP9, ANK2, CACNA1C, CACNA2D1, CACNB2, CALM1, CALM2, CASQ2, CAV3, CSRP3, DES, DPP6, DSC2, DSG2, DSP, GJA5, GPD1L, HCN4, HEY2, IRX3, JUP, KCNA5, KCND3, KCNE1, KCNE2, KCNE3, KCNE5, KCNH2, KCNJ2, KCNJ3,CNJ5, KCNJ8, KCNQ1, LDB3, LMNA, MYBPC3, MYH6, MYH7, MYL2, MYL3, MYL4, MYOZ2, NEXN, PKP2, PLN, RANGRF, RBM20, RYR2, SCN10A, SCN1B, SCN3B, SCN4B, SCN5A, SGCD, SNTA, TAZ, TBX5, TCAP, TGFB3, TMEM43, TNNC1, TNNI3, TNNT2, TPM1, TRDN, TRPM4, TTN, TTR)	Not specified	17 cases of SCD
Yamamoto T. et al. (2019) [31]	Clinical exome	Myotonic dystrophy type 1 (DM1)	A case of SUDY
Grassi S. et al. (2021) [32]	82 genes (ABCC9, ACTC1, ACTN2, AKAP9, ANK2, BAG3, CACNA1C, CACNA2D1, CACNB2, CASQ2, CAV3, CRYAB, CSRP3, DES, DMD, DMPK, DSC2, DSG2, DSP, EMD, FKTN, FLNC, GLA, GPD1L, HCN4, JPH2, JUP, KCND3, KCNE1, KCNE2, KCNE3, KCNE5, KCNH2, KCNJ2, KCNJ5, KCNJ8, KCNQ1, LAMP2, LDB3, LMNA, MYBPC3, MYH6, MYH7, MYL2, MYL3, MYOZ2, MYPN, NEBL, NEXN, NOS1AP, PDLIM3, PKP2, PLN, PRKAG2, RANGRF, RBM20, RYR2, SCN1B, SCN2B, SCN3B, SCN4B, SCN5A, SCN10A, SGCD, SLMAP, SNTA1, TAZ, TCAP, TGFB3, TMEM43, TMPO, TNNC1, TNNI3, TNNT2, TP63, TPM1, TRDN, TRIM63, TRPM4, TTN, TTR, VCL)	Not specified	A case of SUDY
Modena M. et al. (2019) [33]	Whole exome	Not specified	A case of SCD
Shanks G.W. et al. (2018) [34]	99 sudden death-susceptibility genes	Not specified	25 cases of SUDY
Marcondes L. et al. (2018) [35]	Not specified, but the following genes are mentioned: SCN5A, KCNH2, KCNQ1, KCNE2, KCNE1, and KCNJ2	Long QT syndrome (LQTS)	365 cases of SUDY
Jenewein T. et al. (2018) [36]	13 genes (DSC2, DSG2, DSP, HCN4, KCNJ2, KCNQ1, KCNH2, SCN5A, KCNE1, KCNE2, PKP2, RyR2, and SCN4B)	Not specified	A case of SUDY
Neubauer J. et al. (2018) [37]	189 genes (ABCC8, ABCC9, ACAD9, ACADM, ACADS, ACADVL, ACTA2, ACTC1, ACTN2, ACVRL1, ADAMTS10, AGL, AKAP9, ANK2, ANKRD1, ASCL1, ATP5E, BAG3, BMPR1B, BMPR2, BRAF, CACNA1C, CACNA2D1, CACNB2, CALM1, CALM2, CALM3, CALR3, CAMK2G, CASQ2, CAV1, CAV3, CBL, COA5, COL3A1, COL5A1, COL5A2, CPT1A, CPT2, CRYAB, CSRP3, CTF1, CTGF, CTNNA3, DCHS1, DES, DLG1, DMD, DMPK, DNAJC, DNM1L, DOLK, DPP6, DSC2, DSG2, DSP, DTNA, ECE1, EFEMP2, ELN, EMD, ENG, ETFA, ETFB, ETFDH, EYA4, FBN1, FBN2, FHL2, FKRP, FKTN, FLNA, FOXRED1, G6PC, GAA, GATA4, GATA5, GATA6, GATA6, GATAD1, GJA1, GJA5, GJD4, GK, GLA, GLB1, GPD1L, GUSB, HADH, HADHA, HADHB, HCN2, HCN4, HEY2, HFE, HMGCL, HMGCS2, ILK, JPH2, JUP, KCNA5, KCND3, KCNE1, KCNE1L, KCNE2, KCNE3, KCNE5, KCNH2, KCNJ2, KCNJ5, KCNJ8, KCNQ1, LAMA4, LAMP2, LDB3, LMNA, MAP2K1, MAP2K2, MED23, MRPL3, MYBPC3, MYH11, MYH6, MYH7, MYL2, MYL3, MYLK, MYLK2, MYOM1, MYOZ2, MYPN, NEBL, NEXN, Nkx2-5, NOS1AP, NOTCH1, PDLIM3, PKP2, PLN, PRKAG2, PRKG1, PSEN1, PSEN2, RAF1, RANGRF, RBM20, RYR2, SCN10A, SCN1B, SCN2B, SCN3B, SCN4B, SCN5A, SCO2, SDHA, SEMA3A, SGCD, SLC22A5, SLC25A3, SLC37A4, SLMAP, SMAD3, SMAD9, SNTA1, SYNE1, SYNE2, TAZ, TBX5, TCAP, TGFβ2, TGFβ3, TGFβR1, TGFβR2, TMEM43, TMPO, TNNC1, TNNI3, TNNT2, TPM1, TRDN, TRPM4, TRPM7, TSFM, TTN, TTR, VCL, XK, ZASP, and ZNF365)	Not specified	34 cases of SCD
Andersen J.D. et al. (2019) [38]	Whole genome sequencing	Not specified	9 cases of SCD
Raju H. et al. (2019) [39]	6 genes (KCNE1, KCNE2, KCNQ1, KCNH2, SCN5A, and RYR2)	Long QT syndrome (LQTS), Brugada syndrome (BrS), and catecholaminergic polymorphic ventricular tachycardia (CPVT)	197 cases of SUDI, SUDY and SCD (unspecified number)
Graziosi M. et al. (2020) [40]	174 genes (ABCC9, ABCG5, ABCG8, ACTA1, ACTA2, ACTC1, ACTN2, AKAP9, ALMS1, ANK2, ANKRD1, APOA4, APOA5, APOB, APOC2, APOE, BAG3, BRAF, CACNA1C, CACNA2D1, CACNB2, CALM1, CALR3, CASQ2, CAV3, CBL, CBS, CETP, COL3A1, COL5A1, COL5A2, COX15, CREB3L3, CRELD1, CRYAB, CSRP3, CTF1, DES, DMD, DNAJC19 DOLK, DPP6, DSC2, DSG2, DSP, DTNA, EFEMP2, ELN, EMD, EYA4, FBN1, FBN2, FHL1, FHL2, FKRP, FKTN, FXN, GAA, GATAD1, GCKR, GJA5, GLA, GPD1L, GPIHBP1, HADHA, HCN4, HFE, HRAS, HSPB8, ILK, JAG1, JPH2, JUP, KCNA5, KCND3, KCNE1, KCNE2, KCNE3, KCNH2, KCNJ2, KCNJ5, KCNJ8, KCNQ1, KLF10, KRAS, LAMA2, LAMA4, LAMP2, LDB3, LDLR, LDLRAP1, LMF1, LMNA, LPL, LTBP2, MAP2K1, MAP2K2, MIB1, MURC, MYBPC3, MYH11, MYH6, MYH7, MYL2, MYL3, MYLK, MYLK2, MYO6, MYOZ2, MYPN, NEXN, NKX2-5, NODAL, NOTCH1, NPPA, NRAS, PCSK9, PDLIM3, PKP2, PLN, PRDM16, PRKAG2, PRKAR1A, PTPN11, RAF1, RANGRF, RBM20, RYR1, RYR2, SALL4, SCN1B, SCN2B, SCN3B, SCN4B, SCN5A, SCO2, SDHA, SEPN1, SGCB, SGCD, SGCG, SHOC2, SLC25A4, SLC2A10, SMAD3, SMAD4, SNTA1, SOS1, SREBF2, TAZ, TBX20, TBX3, TBX5, TCAP, TGFB2, TGFB3, TGFBR1, TGFBR2, TMEM43, TMPO, TNNC1, TNNI3, TNNT2, TPM1, TRDN, TRIM63, TRPM4, TTN, TTR, TXNRD2, VCL, ZBTB17, ZHX3, and ZIC3)	Arrhythmogenic left ventricular cardiomyopathy (ALVC)	A case of SCD
Simons E. et al. (2021) [41]	61 genes (ABCC9, AKAP9, ANK2, CACNA1C, CACNA2D1, CACNB2, CALM1, CASQ2, CAV3, CTNNA3, DES, DPP6, DSC2, DSG2, DSP, GJA1 (CX43), GJA5 (CX40), GPD1L, HCN4, JUP, KCNA5, KCND3, KCNE1, KCNE2, KCNE3, KCNE5 (KCNE1L), KCNH2, KCNJ2, KCNJ5 (GIRK4), KCNJ8, KCNQ1 (excl. exon 9), LMNA, NKX2-5 (NKX2E), NOS1AP, NPPA, PKP2, PLN, PRKAG2, RANGRF (MOG1), RYR2, SCN1B, SCN2B, SCN3B, SCN4B, SCN5A, SLMAP, SNTA1, TGFB3, TMEM43, TRDN, and TRPM4)	Long QT syndrome type 1 (LQTS1)	A case of SCD
Gaertner-Rommel A. et al. (2019) [42]	174 genes associated with inherited cardiac conditions	Hypertrophic cardiomyopathy (HCM)	A case of SUDY
Mahlke N. et al. (2019) [43]	74 genes associated with inherited cardiovascular conditions	Catecholaminergic polymorphic ventricular tachycardia (CPVT)	A case of SUDY
Neubauer J. et al. (2019) [44]	Whole exome	Not specified	A case of SUDY
Foti F. et al. (2020) [45]	174 genes	Arrhythmic heart disease	A case of SUDY
Manzanilla-Romero H.H. et al. (2023) [46]	48 genes for arrhythmias and then whole exome	Myocarditis	A case of SUDY
Ripoll-Vera et al. (2020) [47]	Between 194 and 380 genes	Not specified	62 cases of SCD

**Table 2 diagnostics-14-01151-t002:** Mutations identified and their associated pathogenicity. HCM: hypertrophic cardiomyopathy; DCM: dilated cardiomyopathy; LQTS: long Q-T syndrome; ARVC: arrhythmogenic right ventricular cardiomyopathy; ARVD: arrhythmogenic right ventricular dysplasia; BrS: Brugada syndrome; CPVT: catecholaminergic polymorphic ventricular tachycardia; LVNC: left ventricular non-compaction.

Mutation	Evaluation	Pathology	Reference
ACTN2: c.355G>A p.(Ala119Thr)	Pathogenic	HCM, DCM	Kraoua L. et al. (2022) [24]
CACNA1C c.2573G>A p.(Arg858His)	Pathogenic	LQTS	Larsen M.K. et al. (2020) [21]
CALR3 c.387dup p.(Ile130Tyrfs*11)	Pathogenic	Familial HCM, ARVC	Neubauer J. et al. (2021) [23]
DSG2 c.2979G>T p.(Gln993His)	Likely pathogenic	ARDV	Simons E. et al. (2021) [41]
KCNE1 c.292C>T p.(Arg98Trp)	Likely pathogenic	LQTS	Marcondes L. et al. (2018) [35]
KCNH2 c.87C>A p.(Phe29Leu)	Pathogenic	LQTS	Larsen M.K. et al. (2020) [21]
KCNH2 c.211G>C p.(Gly71Arg)	Pathogenic	Congenital LQTS	Raju H. et al. (2019) [39]
KCNH2 c.1591C>T p.(Arg531Trp)	Likely pathogenic	LQTS type 2	Scheiper-Welling S. et al. (2022) [24]
KCNH2 c.1600C>T p.(Arg534Cys)	Pathogenic	LQTS	Scheiper-Welling S. et al. (2022) [24]
KCNH2 c.1682C>T p.(Ala561Val)	Pathogenic/likely pathogenic	LQTS	Marcondes L. et al. (2018) [35]
KCNJ2 c.199C>T p.(Arg67Trp)	Pathogenic	LQTS	Marcondes L. et al. (2018) [35]
KCNQ1 c.287C>G p.(Thr96Arg)	Likely pathogenic	LQTS	Marcondes L. et al. (2018) [35]
KCNQ1 c.568C>T p.(Arg190Trp)	Pathogenic/likely pathogenic	LQTS	Larsen M.K. et al. (2020) [21]
KCNQ1 c.727C>T p.(Arg243Cys)	Pathogenic	LQTS	Marcondes L. et al. (2018) [35]
KCNQ1 c.969G>A p.(Trp323Ter)	Pathogenic	LQTS	Raju H. et al. (2019) [39]
KCNQ1 c.1363C>T p.(His455Tyr)	Pathogenic	LQTS	Marcondes L. et al. (2018) [35]
LMNA c.568 C>T p.(Arg190Trp)	Pathogenic	DCM	Marey I. et al. (2020) [15]
LMNA c.1412G>A p.(Arg471His)	Pathogenic/likely pathogenic	DCM	Larsen M.K. et al. (2020) [21]
LMNA c.1579C>T p. (Arg527Cys)	Pathogenic/likely pathogenic	DCM	Neubauer J. et al. (2018) [37]
MYBPC3 c.884delT p.(Phe295fs)	Pathogenic/likely pathogenic	HCM	Siskind T. et al. (2022) [12]
MYBPC3 c.2441_2443del p.(Lys814del)	Likely pathogenic	HCM	Girolami F. et al. (2022) [22]
MYBPC3 c.2670dup p.(Arg891fs)	Likely pathogenic	HCM	Iglesias M. et al. (2021) [20]
MYBPC3 c.2864_2865del p.(PRO955fs)	Pathogenic/likely pathogenic	HCM	Marey I. et al. (2020) [15]
MYBPC3 c.2905C>T p.(Gln969Ter)	Pathogenic	HCM	Siskind T. et al. (2022) [12]
MYH7 c.1325G>A p.(Arg442His)	Pathogenic/likely pathogenic	HCM	Larsen M.K. et al. (2020) [21]
MYH7 c.1955G>A (p.Arg652Lys)	Pathogenic	HCM	Ripoll-Vera T. et al. (2020) [47]
MYH7 c.1988G>A p.(Arg663His)	Pathogenic	HCM	Marey I. et al. (2020) [15]
MYH7 c.2011C>T p.Arg671Cys	Likely pathogenic	HCM	Martínez-Barrios E. et al. (2023) [26]
MYH7 c.2105T>A p.(Ile702Asn)	Pathogenic/likely pathogenic	HCM	Marey I. et al. (2020) [15]
MYH7 c.2155C>T p.(Arg719Trp)	Pathogenic	HCM	Marziliano N. et al. (2019) [18]
MYH7 c.2606G>A p.(Arg869His)	Likely pathogenic	HCM and DCM	Siskind T. et al. (2022) [16]
PKP2 c.235C>T p.(Arg79Ter)	Pathogenic	ARVD	Larsen M.K. et al. (2020) [21]
PKP2 c.1237C>T (p.Arg413Ter)	Pathogenic	ARVD	Martínez-Barrios E. et al. (2023) [26]
PPA2 c.514G>A (p.Glu172Lys)	Pathogenic	PPA2-related mitochondriopathy	Ripoll-Vera T. et al. (2020) [28]
PPA2 c.683C > T p.(Pro228Leu)	Pathogenic	PPA2-related mitochondriopathy	Manzanilla-Romero H.H. et al. (2023) [46]
SCN5A c.127C>T p.(Arg43Ter)	Pathogenic	BrS, LQTS type 3	Scheiper-Welling S. et al. (2022) [24]
SCN5A c.1231G>A p.(Val411Met)	Pathogenic	LQTS type 3	Siskind T. et al. (2022) [16]
SCN5A c.2254G>A p.(Gly752Arg)	Pathogenic/likely pathogenic	BrS	Larsen M.K. et al. (2020) [21]
RYR2 c.1259G>A p.(Arg420Gln)	Pathogenic	LQTS	Raju H. et al. (2019) [39]
RYR2 c.11836G>A p.(Gly3946Ser)	Pathogenic	CPVT	Raju H. et al. (2019) [39]
RYR2 c.13735C>T p.(His4579Tyr)	Likely pathogenic	CPVT	Larsen M.K. et al. (2020) [21]
RYR2 c.13823G>A p.(Arg4608Gln)	Pathogenic/likely pathogenic	CPVT	Raju H. et al. (2019) [39]
RYR2 c.14288A>G p.(Asn4763Ser)	Likely pathogenic	CPVT	Shanks G.W. et al. (2018) [34]
TCAP c.360_361del p.(Glu120Aspfs*15)	Likely pathogenic	HCM	Girolami F. et al. (2022) [22]
TTN c.94344_94347del p.(Lys31448fs)	Likely pathogenic	DCM	Neubauer J. et al. (2018) [37]
TNNI3 c.407G>A p.(Arg136Gln)	Likely pathogenic	HCM	Marey I. et al. (2020) [15]
TNNI3 c.509G>A p.(Arg170Gln)	Pathogenic	HCM	Marey I. et al. (2020) [15]
TNNI3 c.611G>A p.(Arg204His)	Pathogenic/likely pathogenic	HCM	Martínez-Barrios E. et al. (2023) [26]
TNNT2 c.275 G>A p.(Arg92Gln)	Likely pathogenic	HCM and LVNC	Marey I. et al. (2020) [15]
TNNT2 c.421C>T p.(Arg141Trp)	Likely pathogenic	HCM-DCM	Marey I. et al. (2020) [15]
TNNT2 c.517_519del p.(Glu173del)	Pathogenic/Likely pathogenic	HCM	Girolami F. et al. (2022) [22]

## Data Availability

Not applicable.

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
