# Peer review of "New Insights on Molecular Autopsy in Sudden Death: A Systematic Review"

_diagnostics, 2024, doi:10.3390/diagnostics14111151_

Round 1
Reviewer 1 Report
Comments and Suggestions for Authors
In the manuscript, the authors propose the importance of molecular autopsy in forensic pathology identification, especially for the diagnosis of SUD and SIDS. The authors only summarized the results of the published literature; however, no in-depth analysis was conducted. In addition, authors should improve the presentation of results and improve the writing of articles.
All in all, I don't think this article is suitable for publication.
Author Response
“In the manuscript, the authors propose the importance of molecular autopsy in forensic pathology identification, especially for the diagnosis of SUD and SIDS. The authors only summarized the results of the published literature; however, no in-depth analysis was conducted. In addition, authors should improve the presentation of results and improve the writing of articles.
All in all, I don't think this article is suitable for publication.”
- “…The authors only summarized the results of the published literature ; however, no in-depth analysis was conducted …”
Reply: Our work is a systematic review that involves examination of the available literature; we do not believe it to be merely a summary, but rather that an analysis has been conducted. However, we appreciate the reviewer's observation.
-“… authors should improve the presentation of results and improve the writing of articles….”
Reply: the reviewer for their kind observation, we would like to respond that during the revision process, we have corrected the manuscript and made efforts to pay attention to its presentation (as previously done during the initial submission). We hope that these revisions have made it more pleasing to the reviewer.
-“…All in all, I don't think this article is suitable for publication.”
Reply: We thank the reviewer for their opinion and the manner in which it is substantiated. With our revision, we hope to have made the manuscript suitable for publication.

Reviewer 2 Report
Comments and Suggestions for Authors
Dear Editor,
Tomassini et al. provided an interesting manuscript entitled “New insights on molecular autopsy in sudden death: a systematic review” in which the authors performed a systematic review on genetic mutations associated with sudden death.
The manuscript is interesting with possible important clinical applications. The manuscript was correctly performed following PRISMA guidelines.
For readability, could table 1 and 2 be merged? So that it is clear which mutations were found from each gene panel.
Personally, I would have excluded sudden infant death syndrome and sudden death in infants between 1 and 5 years of age, keeping only cases above 5 years old, since the mechanisms are very different in sudden deaths in the very young versus old patients. However, this would imply complete remake of the study, which is not the case. Nevertheless, this is mentioned in the limitations section of the study.
I recommend acceptance of the manuscript.
Author Response
“… Tomassini et al. provided an interesting manuscript entitled “New insights on molecular autopsy in sudden death: a systematic review” in which the authors performed a systematic review on genetic mutations associated with sudden death.
The manuscript is interesting with possible important clinical applications. The manuscript was correctly performed following PRISMA guidelines.
For readability, could table 1 and 2 be merged? So that it is clear which mutations were found from each gene panel.
Personally, I would have excluded sudden infant death syndrome and sudden death in infants between 1 and 5 years of age, keeping only cases above 5 years old, since the mechanisms are very different in sudden deaths in the very young versus old patients. However, this would imply complete remake of the study, which is not the case. Nevertheless, this is mentioned in the limitations section of the study.
I recommend acceptance of the manuscript.”
-“… For readability, could table 1 and 2 be merged? So that it is clear which mutations were found from each gene panel…”
Reply: Thanking the reviewer for their valuable observations, we respond to this point by stating that we attempted several times to merge the tables; however, in our opinion, this significantly compromised the readability of the article, and ultimately, the merging of the tables was not feasible. Nevertheless, we hope to have made the work as accessible as possible.
It should also be noted that in Table 2, multiple rows reference the same author, and in some studies, the genetic panel is not specified, making merging impossible.
-“ …Personally, I would have excluded sudden infant death syndrome and sudden death in infants between 1 and 5 years of age, keeping only cases above 5 years old, since the mechanisms are very different in sudden deaths in the very young versus old patients. . However, this would imply complete remake of the study, which is not the case. Nevertheless, this is mentioned in the limitations section of the study. …”
Reply: We thank the reviewer for their observation and fully share their suggestion; therefore, the revision has been completely modified by including only SUDI and SIDS among the exclusion criteria. These have been considered only in miscellaneous cases where they were included alongside SCD and SUDY because we still deemed the relevant studies essential. However, we have specified this aspect and included it among the limitations.
“…I recommend acceptance of the manuscript:”
Reply: We warmly thank the reviewer for their appreciation; the manuscript has been radically changed in light of their valuable observations.

Reviewer 3 Report
Comments and Suggestions for Authors
Thank you for allowing me the opportunity to review the article. The authors have done a commendable job in exploring the role of molecular autopsy in resolving cases of sudden unexpected death (SUD).
I appreciate the authors for their hard work, but there are certain concerns which need to be addressed:
1. Novelty and relevance: The article addresses an important and emerging area in forensic pathology by highlighting the potential of molecular autopsy to uncover hidden causes of sudden death. However, the novelty of the findings could be better emphasized by comparing them with previous reviews or studies in this field. The authors should clarify how their review advances the understanding of molecular autopsy beyond existing literature.
2. Academic writing, grammar, flow and readability: The writing is generally clear, but there are instances where the language could be polished to improve readability. Some sections are dense and could benefit from simplification for better flow. Additionally, minor grammatical errors need correction to enhance the overall presentation of the manuscript.
3. Narrative structure: While the article follows a logical structure, certain sections could be more cohesive. For instance, the transition between discussing the identification of mutations and the need for standardized post-mortem protocols feels abrupt. A smoother narrative that links the various findings and recommendations would strengthen the manuscript.
4. Methodology: The methodology is well-described, adhering to PRISMA guidelines and providing a clear process for study selection and analysis. Please upload PRISMA checklist as supplementary file.
5. Statistical tools employed: The use of descriptive statistics is appropriate for analyzing the scope of the studies and the mutations identified. However, the manuscript would benefit from a more detailed discussion on the statistical methods used to interpret the results of the reviewed studies and its better to explain reasons for not conducting a meta-analysis (these can be heterogenity of studies, type of data etc.).
6. Discussion: The discussion section effectively highlights the importance of molecular autopsy in cases where standard exams are inconclusive. Nonetheless, it could be enriched by a deeper analysis of the implications of the findings, particularly concerning the identified mutations and their potential impact on forensic practice.
7. Tables and Figures: The tables and figures included in the article are useful in summarizing the data. However, they could be made much better - try making a figure to summarise findings and improve visual appearance of the figure 2. Word/pwerpoint made figures cannot be accepted in this era. Please use the variety of tools available these days.
8. References and literature review: The literature review is thorough, covering relevant studies from the past five years. Yet, the authors should ensure that seminal and foundational studies in the field are also referenced to provide a more complete background. This would strengthen the contextual basis of the review and highlight the progression of research in molecular autopsy.
Overall, the article attempts to add important data, but the present version of the manuscript cannot be considered for publication unless the above concerns are resolved. I hope these comments would help the authors reach a better version of their manuscript.
Comments on the Quality of English LanguageThe academic writing is generally clear and well-structured, but there are instances where the language could be polished to improve readability. Simplifying dense sections and correcting minor grammatical errors would enhance the overall flow and presentation of the manuscript.
Author Response
Response to reviewer
“… Thank you for allowing me the opportunity to review the article. The authors have done a commendable job in exploring the role of molecular autopsy in resolving cases of sudden unexpected death (SUD).
I appreciate the authors for their hard work, but there are certain concerns which need to be addressed:
- Novelty and relevance:The article addresses an important and emerging area in forensic pathology by highlighting the potential of molecular autopsy to uncover hidden causes of sudden death. However, the novelty of the findings could be better emphasized by comparing them with previous reviews or studies in this field. The authors should clarify how their review advances the understanding of molecular autopsy beyond existing literature.
- Academic writing, grammar, flow and readability: The writing is generally clear, but there are instances where the language could be polished to improve readability. Some sections are dense and could benefit from simplification for better flow. Additionally, minor grammatical errors need correction to enhance the overall presentation of the manuscript.
- Narrative structure: While the article follows a logical structure, certain sections could be more cohesive. For instance, the transition between discussing the identification of mutations and the need for standardized post-mortem protocols feels abrupt. A smoother narrative that links the various findings and recommendations would strengthen the manuscript.
- Methodology: The methodology is well-described, adhering to PRISMA guidelines and providing a clear process for study selection and analysis. Please upload PRISMA checklist as supplementary file.
- Statistical tools employed: The use of descriptive statistics is appropriate for analyzing the scope of the studies and the mutations identified. However, the manuscript would benefit from a more detailed discussion on the statistical methods used to interpret the results of the reviewed studies and its better to explain reasons for not conducting a meta-analysis (these can be heterogenity of studies, type of data etc.).
- Discussion: The discussion section effectively highlights the importance of molecular autopsy in cases where standard exams are inconclusive. Nonetheless, it could be enriched by a deeper analysis of the implications of the findings, particularly concerning the identified mutations and their potential impact on forensic practice.
- Tables and Figures: The tables and figures included in the article are useful in summarizing the data. However, they could be made much better - try making a figure to summarise findings and improve visual appearance of the figure 2. Word/pwerpoint made figures cannot be accepted in this era. Please use the variety of tools available these days.
- References and literature review: The literature review is thorough, covering relevant studies from the past five years. Yet, the authors should ensure that seminal and foundational studies in the field are also referenced to provide a more complete background. This would strengthen the contextual basis of the review and highlight the progression of research in molecular autopsy.
Overall, the article attempts to add important data, but the present version of the manuscript cannot be considered for publication unless the above concerns are resolved. I hope these comments would help the authors reach a better version of their manuscript. …”
We extend our gratitude to the reviewer for their observations and for providing us the opportunity to improve our work. Below, we address the individual points raised.
- “… Novelty and relevance:The article addresses an important and emerging area in forensic pathology by highlighting the potential of molecular autopsy to uncover hidden causes of sudden death. However, the novelty of the findings could be better emphasized by comparing them with previous reviews or studies in this field. The authors should clarify how their review advances the understanding of molecular autopsy beyond existing literature. …”
Reply: We thank the reviewers for their valuable observations. Our intention was to provide the work with a focus more on the investigation and study methodology rather than on the causes of sudden death. The clarifications have been inserted into the text in red, along with the relevant citations.
- “… Academic writing, grammar, flow and readability: The writing is generally clear, but there are instances where the language could be polished to improve readability. Some sections are dense and could benefit from simplification for better flow. Additionally, minor grammatical errors need correction to enhance the overall presentation of the manuscript. …”
Reply: We thank the reviewers for their observations. The text has been revised, and efforts have been made to provide some simplifications and correct any typographical errors. Been made additional corrections of citations and errors in the text are.
- “… Narrative structure: While the article follows a logical structure, certain sections could be more cohesive. For instance, the transition between discussing the identification of mutations and the need for standardized post-mortem protocols feels abrupt. A smoother narrative that links the various findings and recommendations would strengthen the manuscript. …”
Reply: The authors thank the reviewers for their observations. Several elements have been added to the text to make it more cohesive, as per their suggestions.
- Methodology: The methodology is well-described, adhering to PRISMA guidelines and providing a clear process for study selection and analysis. Please upload PRISMA checklist as supplementary file.
Reply: We thank the reviewers for their kind observations. The PRISMA checklist has been uploaded as requested.
- Statistical tools employed: The use of descriptive statistics is appropriate for analyzing the scope of the studies and the mutations identified. However, the manuscript would benefit from a more detailed discussion on the statistical methods used to interpret the results of the reviewed studies and its better to explain reasons for not conducting a meta-analysis (these can be heterogenity of studies, type of data etc.).
Reply: We thank the reviewer for their kind observation. In response, we have included a section on the statistical tests employed in the individual studies in the "Results" section. Additionally, at the end of the discussion, we have explained why our review is not suitable for a meta-analysis, as it consists of qualitative studies where statistical results are present in only six works, making a credible comparison unfeasible.
- Discussion: The discussion section effectively highlights the importance of molecular autopsy in cases where standard exams are inconclusive. Nonetheless, it could be enriched by a deeper analysis of the implications of the findings, particularly concerning the identified mutations and their potential impact on forensic practice.
Reply: The reviewers are thanked for their observation. The requested section has been incorporated into the discussion, providing commentary on the results in the context of tangible forensic applications (highlighted in red within the text).
- Tables and Figures: The tables and figures included in the article are useful in summarizing the data. However, they could be made much better - try making a figure to summarise findings and improve visual appearance of the figure 2. Word/pwerpoint made figures cannot be accepted in this era. Please use the variety of tools available these days.
Reply: We thank the reviewers for their observation. We have improved the quality of Figure 2, and an additional summary Figure 3 has been included.
- References and literature review: The literature review is thorough, covering relevant studies from the past five years. Yet, the authors should ensure that seminal and foundational studies in the field are also referenced to provide a more complete background. This would strengthen the contextual basis of the review and highlight the progression of research in molecular autopsy.
Reply: As requested by the reviewers, some more significant studies have been cited in the text. We appreciate the observation.
- Overall, the article attempts to add important data, but the present version of the manuscript cannot be considered for publication unless the above concerns are resolved. I hope these comments would help the authors reach a better version of their manuscript
Reply: The reviewers are thanked for the opportunity to substantially improve the manuscript.
Please see the attachment
